# Evaluation of four regimens of methyl aminolevulinate mediated by red light to treat actinic keratoses: A randomized controlled clinical protocol

Ricardo Hideyoshi Kitamura[1], Milene Alves Azevedo[1], Joao Paulo Ratto Tempestini[2], Christiane Pavani[1], Lara Jansiski Motta[1], Sandra Kalil Bussadori[1,3], Ellen Sayuri Ando-Suguimoto[1], Kristianne Porta Santos Fernandes[1], Raquel Agnelli Mesquita-Ferrari[1,3], Cinthya Cosme Gutierrez Duran[1], Anna Carolina Ratto Tempestini Horliana[1]*

1 Postgraduate Program in Biophotonics-Medicine, Universidade Nove de Julho (UNINOVE), São Paulo, Brazil, 2 Hospital Sírio-Libanês, São Paulo, Brazil, 3 Postgraduate Program in Rehabilitation Sciences, Universidade Nove de Julho (UNINOVE), São Paulo, Brazil

☯ These authors contributed equally to this work.
‡ CP, LJM SKB, ESA-S and CCGD authors also contributed equally to this work.
* annacrth@gmail.com

## Abstract

The multifocality of actinic keratosis (AK), the unpredictability of lesion evolution with potential progression to squamous cell carcinoma (SCC), and the consequent risk of local extension and metastasis, alongside the recent development of new therapies, make the selection of a therapeutic regimen challenging. The increasing incidence of this condition is associated with economic costs and its impact on quality of life, which has fostered interest in studying protocols for treating this skin condition. The topical application of 16% methyl aminolevulinate (MAL) is well-established in the literature for its local therapeutic effects and ease of application. However, the high cost of medication, long incubation time, and adverse effects such as itching and burning in some patients limit the dissemination of this treatment. Studies are needed to test other protocols of this promising therapy to increase acceptance among patients and professionals. Therefore, the objective of this protocol is to compare the efficacy of the topical application of MAL at concentrations of 8% and 16%, mediated by red light, as well as to evaluate the impact of different incubation times (1 or 3 hours) in the treatment of actinic keratoses on the face, with a 6-month follow-up. This parallel-arm, 6-month follow-up randomized controlled, double-blind clinical protocol will consist of 4 groups: G1 - Control Group - MAL 16% irradiated with 643 nm and 75 J/cm² and 3-hour incubation time (n=36), G2 - MAL 16% and 1-hour incubation (n=36), G3 - MAL 8% - 3 hours (n=36), and G4 - MAL 8% - 1 hour (n=36). The primary outcome will be the complete remission of the lesion at six months. Secondary outcomes will include treatment success (75% reduction in the initial number of lesions), recurrence rate, emergence of SCC, incidence of adverse effects, and improvement in skin texture, wrinkles, and pigmentation using a validated scale. All outcomes will be assessed at 30 days, 3, and 6 months. At six months, quality of life will be

**Data availability statement:** Deidentified research data supporting this study's findings have been made publicly available on the Open Science Framework (OSF) at https://osf.io/twrcn/(DOI10.17605/OSF.IO/TWRCN).

**Funding:** Ricardo Hideyoshi Kitamura receives a governmental grant from the Coordination for the Improvement of Higher Education Personnel (CAPES), process number 88887.841293/2023-00. Anna Carolina Ratto Tempestini Horliana also receives support from a governmental grant provided by the National Council for Scientific and Technological Development (CNPq), process number 316287/2023-7.

**Competing interests:** The authors have declared that no competing interests exist.

assessed using the Actinic Keratosis Quality of Life questionnaire (AKQoL) and Face-Q. If data are normal, they will be subjected to 3-way ANOVA and presented as means ± standard deviation (SD). Otherwise, they will be presented as median and interquartile range and compared using the Kruskall-Wallis and Friedman tests. Categorical variables will be evaluated with the chi-square, Fisher's exact, or likelihood ratio tests. A p-value $< 0.05$ will be considered significant.

## Introduction

Actinic keratoses (AK) account for 10% of dermatological consultations in Brazil [1]. The statistic of 18.8% refers to the proportion of participants who developed squamous cell carcinoma (SCC) over 10 years. Additionally, this study included 2,893 patients with AK, who were followed during this timeframe and compared with a control group without AK. The findings revealed that patients with AK had a more than fivefold increased risk of developing skin cancer overall. Specifically, the risk was notably higher for SCC, with patients demonstrating a more than sevenfold increased risk [2].

Additionally, there is a significant impact on the quality of life of individuals as the lesions cause pain and bleeding in photo-exposed areas, leading to limitations and promoting appearance-related stigmas that interfere with social interaction, profession, leisure, and self-esteem [3]. Therefore, there is great interest in discovering therapeutic alternatives, with the topical application of MAL followed by PDT irradiation considered one of the most promising, as it is less destructive and more selective and can also be used to treat pre-cancerous fields [4–9]. The higher selectivity of porphyrin precursors for diseased cells is crucial for the superior aesthetic results of this method compared to others. However, it can also cause discomfort, pain, and burning for a few minutes during photoactivation and for a few hours afterward due to the inflammatory reaction, approximately two hours for MAL and six hours for 5-ALA [10,11]. It highlights the need to develop protocols to mitigate adverse events. The multifocality of actinic keratosis, the unpredictability of lesion evolution with possible progression to SCC, and the consequent risk of local extension and metastasis, along with the recent development of new therapies, make the selection of a therapeutic regimen a challenging task [12–16]. Furthermore, the increasing incidence, associated economic costs, and impact on quality of life have fostered interest in reviewing protocols for treating this skin condition. The topical application of 16% MAL is well established in the literature for its local therapeutic effects and ease of application [17,18]. However, the high cost of medication, long incubation time, and adverse effects such as itching and burning limit the dissemination of this treatment. Studies are needed to test other protocols of this promising therapy to increase acceptance among patients and professionals [19]. Lower concentrations could make the product more accessible with fewer adverse effects [20], but more studies are needed to increase the level of evidence. Moreover, the 3-hour incubation time discourages professionals and patients from choosing this therapeutic modality for treating this condition. The objective of this study is to compare the efficacy of the topical application of MAL at concentrations of 8% and 16%, mediated by red light, as well as to evaluate the impact of different incubation times (1 or 3 hours) in the treatment of actinic keratoses on the face, with a 6-month follow-up.

## Materials and methods

This will be a randomized, controlled, parallel-group, prospective clinical protocol following the SPIRIT Statement clinical protocol criteria (S1 Appendix). This protocol was approved on 20th March 2024 by the Ethics Committee (process number 6.715.052). We included one

more outcome (Face-Q analysis) on 4th October, before the start of the study, and it was also approved by the Ethics Committee (process number 7.122.841). The entire protocol was also presented as submitted to the Ethics Committee (S2 Appendix and S3 Appendix). The principal investigator will individually recruit and invite patients awaiting treatment to participate at the medical and dental outpatient clinics of Universidade Nove de Julho in São Paulo, Brazil. Additionally, individuals attending the private practice of the principal investigator, Clínica Médica Perfecthaderm, located at Alameda Santa Cruz, 525 – Adamantina – SP, Brazil, will also be invited to participate (S4 Appendix) in Clinicaltrials.gov (NCT06507644 approved on 11th July 2024, updated on 09th October 2024, with Face Q inclusion). Participants who meet the inclusion criteria will be invited to the clinical outpatient clinic at UNINOVE, São Paulo, Brazil, for dermatological treatment by the principal investigator (dermatologist) from 30th November 2024 to 30th November 2025, with a final 6-month follow-up in 30th April 2026. Those who agree to participate will sign the Informed Consent (IC) after receiving a detailed verbal and written explanation from the principal investigator.

## Sample size calculation

Considering Braathen et al.'s 2008 study, we observed an average variation of 64% to 91% complete response to the lesion after 3 months in the 8% (1 and 3 hours) and 16% (1 and 3 hours) groups. With 95% confidence and 80% power to detect differences between the groups, the minimum sample size will be 36 participants per group, totaling 144 participants, calculated using a chi-square test for proportions.

## Strategies for achieving adequate participant enrolment

It will include promoting the study in the city to reach the target sample size through social media.

## Calibration and training of evaluator

The principal investigator (dermatologist-researcher 1) will collect all study outcomes; therefore, only this researcher will be calibrated. He will be responsible for evaluating 5 participants with facial actinic keratosis lesions. Each of these 5 participants will be evaluated (T0), and the lesions will be quantified. The 5 participants will be seen in sequence and again after 1 hour. The same evaluation (recounting of lesions) (T1) will be performed, and the results will be recorded. The Intraclass Correlation Coefficient (ICC) will be calculated to assess the intra-examiner agreement of T0 and T1 values. A value ≥ 0.80 will be considered adequate regarding lesion count agreement. These procedures are essential to maximize the reproducibility of the evaluations. These participants will receive clinical dermatological treatment for actinic keratoses according to their needs. These evaluations will not be part of the study but will serve to certify the intra-examiner agreement.

## Sample description

The sample will consist of individuals with photodamaged skin affected by multiple facial actinic keratoses of grade I (thin), grade II (moderately thick), or grade III (thick), as defined by [21].

  **Inclusion criteria:**

- Individuals of both sexes,

- Aged between 40 and 90 years,

- Fitzpatrick skin phototypes I to IV,

- Photodamaged skin with at least five clinically evident actinic keratosis lesions on the face to be treated,

- No prior treatment for at least six months.

   **Exclusion criteria:**

- Clinically diagnosed infiltrative lesions, as the gold standard treatment is surgical with histopathological evaluation of the lesion (surgery will be performed at no cost to the participant), who will receive guidance and referral for appropriate treatment,

- Photosensitive diseases, such as systemic lupus erythematosus, dermatomyositis, and porphyria,

- History of arsenic exposure,

- Known allergy to MAL or similar photosensitizing agents,

- Psychoactive drug abuse,

- Previous radiotherapy at the lesion site(s),

- Participation in another clinical trial,

- Intense tanning at the time of treatment,

- Pregnant or breastfeeding women,

- Local or systemic infection,

- Immunosuppression: uncompensated chronic diseases or emotional disorders considered contraindications to treatment,

- Skin conditions on the neck and anterior chest.

## Randomization

To randomly allocate participants into the experimental groups, 144 numbers will be drawn using the Excel program. The group distribution will be identical (1:1:1:1) for groups, with block randomization (24 blocks of 6 participants). Opaque envelopes will be identified with sequential numbers containing the corresponding experimental group information. The envelopes will be sealed and remain closed in numerical order until the time of lesion treatment. The draw and envelope preparation will be performed by a person not involved in the study (ACRTH). Immediately before the lesion treatment, the researcher responsible for the treatment will open the envelope (without altering the numerical sequence) and perform the indicated procedure. RHK (research 1) will enroll participants and obtain informed consent. A separate researcher (researcher 2) will handle participant assignment to interventions to maintain confidentiality.

## Blinding

Only the researcher responsible for performing the treatments (who will open the randomization envelopes) will know which treatment is assigned to each participant (Research 2). The group identification will be revealed only after statistical data analysis to all involved in the study. Therefore, the researcher responsible for data collection (researcher 1) and their assistant will be blinded to the treatments assigned to the groups. The participant and the statistician will also be blinded.

## Pre-treatment evaluations

Participants with actinic keratosis who sign the TCLE will undergo anamnesis and complete the AKQoL questionnaire and Face-Q. The principal investigator (calibrated) will count, classify, and map the facial lesions. Photographic records of the lesions will also be taken with a 3D Quantificare® camera and an iPhone 11 Pro Max camera.

## Anamnesis

Anamnesis will be conducted with participants from all groups. Besides questions related to the participant's general health, demographic data (age, gender, marital status, occupation, educational level, living conditions, family income) and medical history data (main complaint, current disease status, medical history, medications) will be collected.

## Experimental design

Immediately before lesion treatment, the researcher will remove and open one envelope (without altering the numerical sequence of the remaining envelopes) and perform the indicated procedure. Thus, individuals will be allocated to the experimental groups as follows (Fig 1):

| | RECORD | RANDOMIZATION | INTERVENTIONS | STUDY PERIOD POS OPERATIVE | | CLOSURE |
|---|---|---|---|---|---|---|
| **TIME** | $-T_1$ | 0 | | $T_1$ (30 d) | $T_1$ (3 months) | $T_3$ (6 months) |
| **RECORD:** | | | | | | |
| Inclusion/exclusion criteria | X | | | | | |
| Informed consent | X | | | | | |
| Research calibration | X | | | | | |
| Medical history | X | | | | | |
| Assignment | | X | | | | |
| **INTERVENTIONS:** | | | | | | |
| G1: Control Group MAL 16% - 3-hour incubation | | | X | | | |
| G2: Experimental Group MAL 16% - 1-hour incubation | | | X | | | |
| G3: Experimental Group MAL 8% - 3-hour incubation | | | X | | | |
| G4: Experimental Group MAL 8% - 1-hour incubation | | | X | | | |
| **OUTCOMES:** | | | | | | |
| Photographic documentation of lesions | X | | | X | X | X |
| Initial lesion count | X | | | X | X | X |
| Initial assessment of lesion signs and symptoms (if present) | X | | | X | X | X |
| Evaluation of skin texture | X | | | X | X | X |
| Quality of life questionnaire, | X | | | | | X |
| Face-Q | X | | | | | X |
| | | | | | | |
| | | | | | | |
| | | | | | | |

**Fig 1. SPIRIT flowchart.**

G1 - Control Group (gold standard – 16% MAL with 3-hour incubation time) (n = 36): Participants will be treated with 16% topical MAL photosensitizer (Metvix®, Galderma, ANVISA registration number 1291600650016) with a 3-hour incubation period. The light source used for skin illumination will be a visible light source (LED) with a wavelength of 643 nm (Hygialux LLT1601®, KLD - ANVISA registration number 10245239012).

G2 - Experimental Group (optimized time using the gold standard medication 16% MAL with shorter incubation time - 1 hour) (n = 36): Participants will be treated with 16% topical MAL photosensitizer (Metvix®, Galderma, ANVISA registration number 25351.002042/2004-68) with a 1-hour incubation period. The light source used for skin illumination will be a visible light source (LED) with a wavelength of 643 nm (Hygialux LLT1601®, KLD - ANVISA registration number 10245239012).

G3 - Experimental Group (compounded medication with lower concentration – 8% MAL with conventional 3-hour incubation time) (n = 36): Participants will be treated with 8% topical MAL photosensitizer (compounded by StinPharma – Industrial Standard Compounding Pharmacy) with a 3-hour incubation period. The light source used for skin illumination will be a visible light source (LED) with a wavelength of 643 nm (Hygialux LLT1601®, KLD - ANVISA registration number 10245239012).

G4 - Experimental Group (compounded medication with lower concentration – 8% MAL with a shorter incubation time of 1 hour) (n = 36): Participants will be treated with 8% topical MAL photosensitizer (compounded by StinPharma – Industrial Standard Compounding Pharmacy) with a 1-hour incubation period. The light source used for skin illumination will be a visible light source (LED) with a wavelength of 643 nm (Hygialux LLT1601®, KLD - ANVISA registration number 10245239012). To improve adherence to intervention protocols, the principal researcher will provide sunscreen.

## Topical photosensitizer MAL treatment

Before the treatment, the treated area will be degreased with 0.2% aqueous chlorhexidine. Then, the dermatologist will perform a light curettage in all the AK lesions (this procedure specifically targets all the AK lesions, not the entire facial area) on the face with a sterile curette. The curettage is meticulously performed to ensure that only the areas with AK are treated, thereby optimizing the effectiveness of the subsequent photodynamic therapy. After curettage, a thin layer of the photosensitizing medication, approximately 1 mm thick, will be applied to the participant's facial lesion sites. Then, an occlusive dressing will be used to enhance MAL penetration, which will be covered with aluminum foil to prevent ambient light from influencing the protoporphyrin production process.

## Conventional PDT protocol

For the conventional PDT technique, the dressing will remain on the face for one hour for participants in G2 and G4 and three hours for participants in G1 and G3. Participants who request it will be allowed to return home and return to the clinic after the required period (1 or 3 hours, depending on the protocol), thus being treated on the same day.

After the proposed incubation period, the dressing will be removed, and the excess photosensitizing medication will be cleaned with gauze soaked in 0.9% saline solution before exposure to light. The distance between the red LED lamp (Light Emitting Diode) device, with a narrow action spectrum, and the skin will be approximately 1 cm. The LED will be the Hygialux LLT1601®, produced by KLD, São Paulo, Brazil. The parameters used were properly calculated (Table 1).

**Table 1. Dosimetric parameters.**

| Parameter | Value |
|---|---|
| Central wavelength (nm) | 643 |
| Spectral width (FWHM) (nm) | 20 |
| Mode of operation | Continuous |
| Average radiant power per LED (mW) | 19 |
| Average radiant power (mW) | 22169.2 |
| Polarization | Random |
| Beam profile | Multimode |
| Beam size on target ($cm^2$) | 465.6 |
| Irradiance on target ($mW/cm^2$) | 48 |
| Exposure time (s) | 1570 |
| Radiant exposure on target ($J/cm^2$) | 75 |
| Radiant energy per session (J) | 34805.6 |
| Application technique | 1 cm distance from the target |
| Session frequency | 1 |
| Number of sessions | 1 |
| Total radiant energy (J) | 34805.6 |
| Photosensitizer | MAL 8% and MAL 16% |
| Pre-irradiation time | 1 h or 3 h |

nm (nanometers), FWHM (Full Width at Half Maximum), mW (milliwatts), $cm^2$ (square centimeters), $mW/cm^2$ (milliwatts per square centimeter), s (seconds), J (joules), $J/cm^2$ (joules per square centimeter), cm (centimeters), h (hours).

Skin illumination will be performed using a visible light source (LED) with a wavelength of 643 nm (ANVISA Registration 10245239012).

According to the manufacturer's instructions, the energy amount per session is 75 $J/cm^2$ (Metvix® package insert). This is considered the standard energy amount in the literature [22]. A physicist specializing in this area (Dr. Alessandro Deana) based all other calculations on this standard dose. After the session, participants will be instructed to avoid sun exposure for one week and prescribed a sun protection lotion with SPF 30. Free samples will be provided to the participants.

## Medication

All participants will receive a medicinal prescription in case of pain:

- Paracetamol every 6 hours for 3 days, only if there is pain. They will be instructed to contact the responsible physician researcher via cell phone, and further prescriptions will be provided if needed. All participants will leave the consultation with the medication prescription but only take it if necessary.

- Toragesic® (Ketorolac Tromethamine) 10 mg every 6 hours (maximum dose for elderly over 65 - 40 mg/day).

**The outcomes of the study will be** (Fig 2)**:**

**Complete remission.** Complete remission will be the study's primary outcome (Quantitative evaluation): A count of the number of lesions with complete response, i.e., those that show total disappearance measurable after treatment, will be performed. These lesions will be clinically evaluated at 30 days, 3 months, and 6 months post-treatment, and

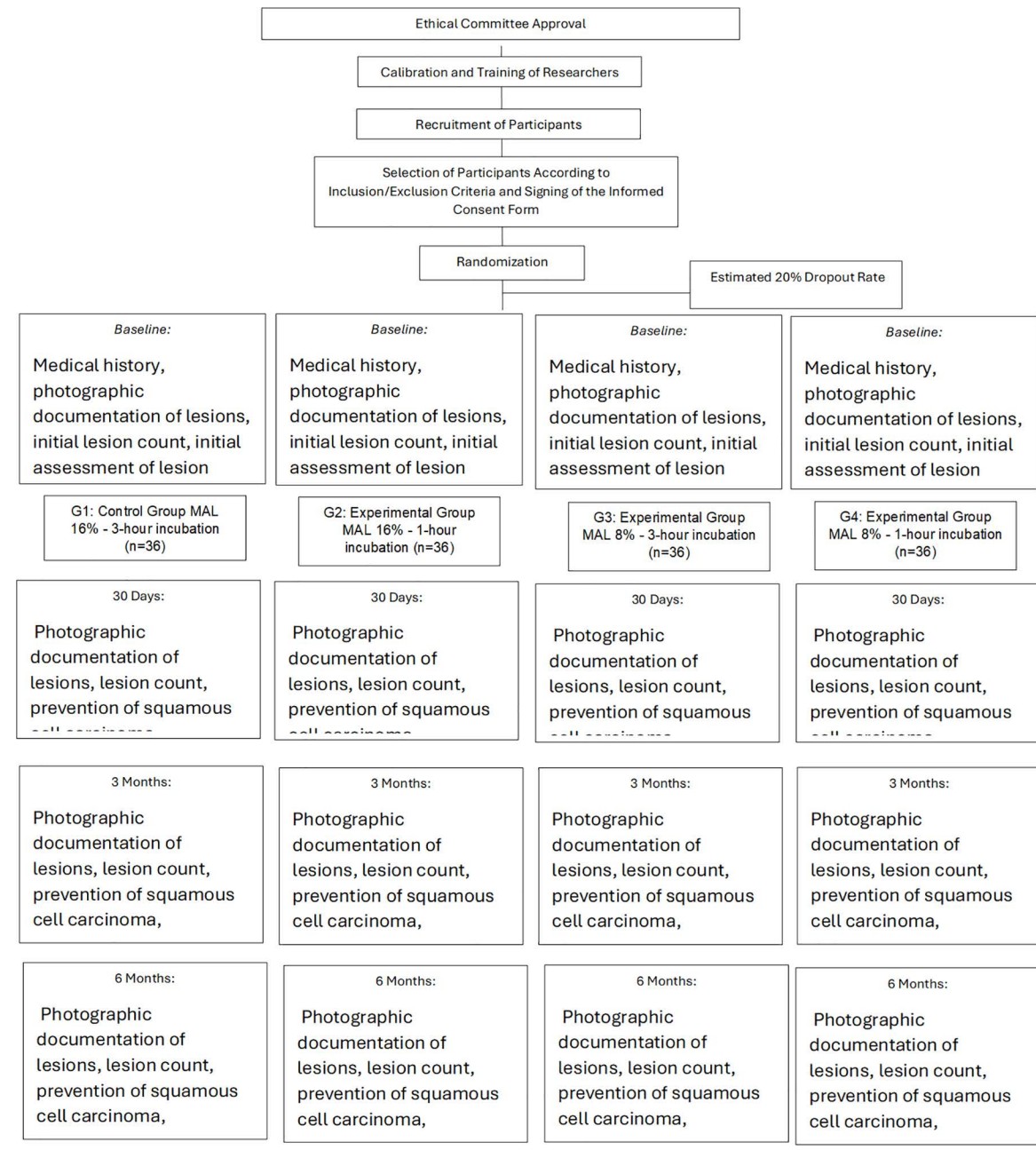

**Fig 2. Study flowchart.**

the number of lesions at these periods will be compared with the initial value (baseline). Both the absolute and relative number of lesions will be considered. To avoid variability in counting, only one researcher will perform the counts. Complete remission of the lesions will be considered when it occurs in 100% of the lesions after 6 months of treatment, as described by others [23].

The secondary outcome variables of the study will be:

**Treatment success.** Evaluation of the proportion of participants who show at least a 75% reduction in the initial number of actinic keratosis lesions in the treatment area after the last

day of treatment. Evaluations will be performed at 30 days, 3 months, and 6 months. Both the absolute and relative number of lesions will be considered. Treatment success will be defined as complete remission observed in at least 75% of participants over 6 months.

**Actinic keratoses recurrence rate.** Defined as the reappearance of lesions in previously treated areas. Recurrence evaluation will occur at the same follow-up periods of the study, i.e., at 30 days, 3 months, and 6 months after treatment. Recurrent lesions will be quantified, considering the absolute and relative number of lesions. These lesions will be monitored and re-treated at the end of the study unless malignization occurs, in which case they will be treated immediately.

**Prevention of squamous cell carcinoma (SCC).** If malignization of the lesion occurs during the follow-up period, the gold standard treatment, which consists of surgical intervention, will be applied. Participants will be continuously monitored to prevent the development of squamous cell carcinoma in the treatment area throughout the study. The quantification of malignant lesions will be done, considering the absolute and relative numbers.

**Incidence of adverse effects.** The incidence of adverse effects, such as erythema, edema, itching, and peeling, will be monitored through a personal diary filled out by the participant, in which detailed descriptions of any adverse effects will be recorded. As recommended by some authors [23], this method will allow participants to report their symptoms individually. The responsible researcher, a dermatologist specializing in this type of treatment, will offer continuous assistance and follow-up, remaining accessible whenever necessary.

**Subjective pain assessment.** Subjective pain assessment will be conducted using the Visual Analog Scale (VAS), consisting of a 10 mm line with closed ends, indicating '0' for no pain and '10' for unbearable pain, the worst pain ever felt. The same operator will consistently provide instructions for marking. Each participant will be instructed to mark the point that best reflects the intensity of the pain at the time of evaluation with a vertical line [24,25]. These evaluations will be performed weekly until 30 days after treatment, followed by assessments at 3 and 6 months.

**Rescue medication.** Rescue medication will be evaluated by the standardized number of analgesics ingested (paracetamol). At the beginning of the study, each participant will receive a blister pack of paracetamol®, a drug with a purely analgesic effect [26]. Participants are instructed to keep the blister pack until the end of the experiment and bring it to each consultation. At the end of the study, the number of tablets used will be evaluated in each group as a parameter for measuring pain.

**Evaluation of skin texture, wrinkles, and pigmentation.** This will be conducted at 30 days, 3 months, and 6 months, using the Tina Alster et al. (2005) scale [27]. This scale, evaluated by professionals and participants themselves, classifies improvements as minimal (<25%), moderate (25%–50%), significant (51%–75%), and excellent (>75%). These evaluations will provide a comprehensive approach to measure the treatment's effectiveness over time.

**Participant satisfaction.** After 6 months of treatment, the Actinic Keratosis Quality of Life questionnaire (AKQoL) [28] will assess participant satisfaction. The items will be scored on a standard 4-point Likert scale and summarized into a maximum total score of 32 points. A higher score indicates greater impairment in quality of life. The questionnaire has been translated and validated into Portuguese [29] (Table 2).

**Evaluation of satisfaction with facial appearance (FACE-Q).** The FACE-Q scale (Satisfaction with Facial Appearance Overall) was developed by Klassen [30] and consists of 10 items to be answered using a four-point Likert scale, which measures satisfaction with facial appearance in various scenarios. A Likert scale consists of a set of items in which the

subject being evaluated is asked to indicate their level of agreement, ranging from "strongly disagree" (level 1) to "strongly agree" (level 4). The FACE-Q evaluates the perception of facial appearance in terms of characteristics such as symmetry, harmony, proportion, freshness or vitality, temporal appearance (e.g., rested facial appearance at the end of the day or upon waking), appearance under brighter lighting, appearance in photographs, and profile (lateral view or contour of the face). Despite being recent, several studies have already used the scale [31]–[33]. Gama carried out its adaptation to Brazilian Portuguese in 2018 [34]. The FACE-Q scale, validated for use in Brazil (Table 3), consists of nine items that measure two satisfaction factors (Overall facial appearance and Facial geometry). The item "With the appearance of your face at the end of the day" from the original scale was removed in the Brazilian version due to its low representativeness in the Brazilian sample. For calculation purposes, it was also excluded from the evaluation in the present study. The sum of the scores obtained from patients' responses to the nine items (1 = very dissatisfied, 2 = somewhat dissatisfied, 3 = somewhat satisfied, and 4 = very satisfied) can range from 9 to 36 and are converted into a score ranging from zero to 100. Higher scores indicate greater satisfaction (Table 4). The participants answered the FACE-Q questionnaire adapted to Brazilian Portuguese before treatment and 30 days after completion.

## Criteria for discontinuing or modifying allocated interventions

If, during the consultations, there were some clinically diagnosed infiltrative lesions, as the gold standard treatment is surgical with histopathological evaluation of the lesion (surgery

**Table 2. Actinic Keratosis Quality of Life questionnaire (AKQoL).**

| During the last week, I: | Not at all | A little bit | Very much | Extremely |
|---|---|---|---|---|
| felt bothered by having to protect my skin every time I go out in the sun. | | | | |
| my sun-damaged skin made me think about what is important in life. | | | | |
| my quality of life suffered due to my sun-damaged skin. | | | | |
| was afraid that my sun-damaged skin might develop into a more serious skin disease. | | | | |
| tried to hide the sun damage on my skin from others with makeup or clothes. | Rarely/Not at all | Sometimes | Frequently | Constantly |
| felt guilty about the sun's damage to my skin. | Rarely/Not at all | Sometimes | Frequently | Constantly |
| observed my skin and checked for sun damage. | Rarely/Not at all | Sometimes | Frequently | Constantly |
| found my life more difficult because of the sun damage to my skin. | | | | |
| thought about how to behave when exposed to the sun. | Rarely/Not at all | Sometimes | Frequently | Constantly |

**Table 3. Items that compose the Face-Q evaluation. Item d) was removed in the adaptation for use in Brazil.**

**CONSIDERING YOUR FACE AS A WHOLE, OVER THE LAST WEEK, HOW SATISFIED OR DISSATISFIED HAVE YOU BEEN WITH EACH ITEM BELOW:**

ITEMS

a) With the symmetry of your face (how similar it looks on both sides)?
b) With the harmony of your face?
c) What is the proportion of your face?
d) With the appearance of your face at the end of the day?*
e) how fresh does your face look?
f) With how rested your face look does?
g) With the appearance of your profile (side view)?
h) What About the appearance of your face in photos?
i) What is the appearance of your face when you wake up?
j) With the appearance of your face under intense (or strong) lighting?

**Table 4. Conversion of raw score to a 0 to 100 scale, according to the Brazilian version of the Face-Q.**

| CONVERSION - TOTAL SATISFACTION | |
|---|---|
| RAW SCORE | RASCH SCORE EQUIVALENT (0–100 scale) |
| 9 | 0 |
| 10 | 4 |
| 11 | 7 |
| 12 | 11 |
| 13 | 15 |
| 14 | 19 |
| 15 | 22 |
| 16 | 26 |
| 17 | 30 |
| 18 | 33 |
| 19 | 37 |
| 20 | 41 |
| 21 | 44 |
| 22 | 48 |
| 23 | 52 |
| 24 | 56 |
| 25 | 59 |
| 26 | 63 |
| 27 | 67 |
| 28 | 70 |
| 29 | 74 |
| 30 | 78 |
| 31 | 81 |
| 32 | 85 |
| 33 | 89 |
| 34 | 93 |
| 35 | 96 |
| 36 | 100 |

**Source**: Gama (2018) [34].

will be performed at no cost to the participant), they will receive guidance and referral for appropriate treatment.

## Analysis of results

Initial descriptive analyses will consider all study variables, quantitative (mean and standard deviation) and qualitative (frequencies and percentages). If the data are normal, a 3-way ANOVA will be performed, and the data will be presented as means ± standard deviation (SD). Otherwise, they will be presented as medians and interquartile ranges and compared using the Kruskal-Wallis and Friedman tests for time comparisons. Categorical variables will be evaluated using the chi-square, Fisher's exact, or likelihood ratio tests. Subgroup or adjusted analyses can be done depending on the variability of lesions (quantity) In all tests, a significance level of 5% probability or the corresponding p-value will be adopted. All analyses will be conducted using the SAS for Windows statistical software, version 9.1. We will use intention-to-treat analysis combined with multiple imputation

methods if the data loss exceeds 10%, as this level of missing data warrants a more robust approach to ensure the validity of our results. We will use intention-to-treat analysis combined with multiple imputation methods if the data loss exceeds 10%, as this level of missing data warrants a more robust approach to ensure the validity of our results. No interim analyses or stopping guidelines are planned, as the interventions are deemed safe. We will collect, share, and maintain the personal information of potential and enrolled participants with strict confidentiality (only among researchers) protocols before, during, and after the trial.

## Plans for communicating important protocol modifications

Researchers will communicate changes to eligibility criteria, outcomes, and analyses to relevant parties—including investigators, IRBs, trial participants, trial registries, and journals, as outlined to ensure transparency and adherence to ethical and regulatory standards.

## Discussion

This study protocol outlines the design for evaluating the efficacy of two different concentrations of methyl aminolevulinate (MAL), 8% and 16%, with varying incubation times (1 and 3 hours) for the treatment of facial actinic keratoses (AK). The rationale behind this study is grounded in the need to optimize photodynamic therapy (PDT) protocols, balance treatment effectiveness with patient comfort, and minimize adverse effects.

Previous studies have demonstrated that higher concentrations of MAL (e.g., 16%) and longer incubation times tend to result in better lesion clearance [20]. However, these benefits are often accompanied by increased patient discomfort, such as pain, erythema, and post-treatment irritation [10]. This study explores whether using a lower concentration of 8% MAL combined with shorter incubation times can still deliver effective treatment outcomes while potentially minimizing these side effects. A secondary aim of the study is to evaluate the recurrence rate of AK lesions following treatment. While existing literature suggests that higher MAL concentrations are associated with lower recurrence rates [17], it remains to be seen whether the proposed lower-concentration protocol can offer a comparable long-term benefit. Understanding this aspect is crucial for developing a treatment that is both efficient and sustainable over time.

Moreover, this protocol innovates by including aesthetic outcomes, such as improved skin texture and pigmentation. Photodynamic therapy has been shown to offer cosmetic benefits in addition to its therapeutic role in AK treatment [27]. The current study will assess these aesthetic outcomes, which could be important in-patient satisfaction and treatment acceptance.

This study's design will also address potential adverse effects. By comparing different MAL concentrations and incubation times, we aim to gather data on how these variables impact both treatment efficacy and patient comfort. This approach is consistent with previous findings that suggest shorter incubation times may reduce pain and discomfort without compromising efficacy [19].

In summary, this protocol presents a comprehensive approach to assessing the efficacy of MAL-PDT with different concentrations and incubation times for treating facial AK. This study's results can potentially inform clinical practice by identifying a protocol that optimally balances efficacy, safety, and patient satisfaction. This could lead to more personalized treatment options, improving outcomes for individuals with AK.

**Monitoring.** The investigators do not plan interim analysis, as no serious adverse events are expected. However, all adverse events will be recorded.

## Supporting information

**S1 Appendix. Appendix spirit.**
(DOC)

**S2 Appendix. Study protocol approved by IRB in English.**
(DOCX)

**S3 Appendix. Study protocol approved by IRB in original language.**
(DOCX)

**S4 Appendix. Clinicaltrials.**
(PDF)

## Author contributions

**Conceptualization:** Ricardo Hideyoshi Kitamura, Christiane Pavani, Raquel Agnelli Mesquita-Ferrari, Anna Carolina Ratto Tempestini Horliana.

**Data curation:** Ricardo Hideyoshi Kitamura, Milene Alves Azevedo, Joao Paulo Ratto Tempestini.

**Formal analysis:** Lara Jansiski Motta, Ellen Sayuri Ando-Suguimoto, Cinthya Cosme Gutierrez Duran.

**Project administration:** Ricardo Hideyoshi Kitamura, Lara Jansiski Motta, Sandra Kalil Bussadori, Kristianne Porta Santos Fernandes, Anna Carolina Ratto Tempestini Horliana.

**Validation:** Christiane Pavani, Sandra Kalil Bussadori, Kristianne Porta Santos Fernandes, Raquel Agnelli Mesquita-Ferrari, Cinthya Cosme Gutierrez Duran, Anna Carolina Ratto Tempestini Horliana.

**Writing – original draft:** Ricardo Hideyoshi Kitamura, Milene Alves Azevedo, Joao Paulo Ratto Tempestini, Ellen Sayuri Ando-Suguimoto.

**Writing – review & editing:** Kristianne Porta Santos Fernandes, Raquel Agnelli Mesquita-Ferrari, Cinthya Cosme Gutierrez Duran, Anna Carolina Ratto Tempestini Horliana.

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
