## [Decision Letter · Decision Letter 0]

10 Dec 2024

PONE-D-24-48146Evaluation of four regimens of methyl aminolevulinate mediated by red light to treat actinic keratoses: a randomized controlled clinical protocolPLOS ONE

Dear Dr. Horliana,

Thank you for submitting your manuscript to PLOS ONE. After careful consideration, we feel that it has merit but does not fully meet PLOS ONE’s publication criteria as it currently stands. Therefore, we invite you to submit a revised version of the manuscript that addresses the points raised during the review process.

We look forward to receiving your revised manuscript.

Kind regards,

Liang Liu

Academic Editor

PLOS ONE

4. We note that the original protocol that you have uploaded as a Supporting Information file contains an institutional logo. As this logo is likely copyrighted, we ask that you please remove it from this file and upload an updated version upon resubmission.

Reviewers' comments:

Reviewer's Responses to Questions

**Comments to the Author**

1. Does the manuscript provide a valid rationale for the proposed study, with clearly identified and justified research questions?

Reviewer #1: Yes

Reviewer #2: Yes

2. Is the protocol technically sound and planned in a manner that will lead to a meaningful outcome and allow testing the stated hypotheses?

Reviewer #1: Yes

Reviewer #2: Yes

3. Is the methodology feasible and described in sufficient detail to allow the work to be replicable?

Reviewer #1: Yes

Reviewer #2: Yes

4. Have the authors described where all data underlying the findings will be made available when the study is complete?

Reviewer #1: No

Reviewer #2: Yes

5. Is the manuscript presented in an intelligible fashion and written in standard English?

Reviewer #1: Yes

Reviewer #2: Yes

6. Review Comments to the Author

You may also provide optional suggestions and comments to authors that they might find helpful in planning their study.

Reviewer #1: Overall, this protocol is well written and clear in the proposed methodology. While similar methodologies have been used previously, testing the reproducibility of previous trials will be meaningful. Additionally, the inclusion of secondary outcomes not previously tested will also be of interest to clinicians and researchers. Areas for minor revision are below.

Line 61 - The statement "Although the annual risk of progression of this type of lesion to squamous cell carcinoma (SCC) is 0.025%- 0.6%, participants with multiple lesions have a risk of up to 20% for the emergence of carcinoma

[2]" should be rephrased. As it is written, it may be interpreted that patients with multiple AK have a 20% risk of emergence of SCC from AK. The statistic of 18.8% was for the number of participants developing squamous cell carcinoma in general, not necessarily for developing squamous cell carcinoma from a pre-existing AK. Additionally, this statistic was measured over a 10 year period.

Line 221 - It would be beneficial to explain how areas for curettage will be chosen and to clarify that curettage will be done over actinic keratoses and not over the entire face (unless this is not the case and if so this should be made clear).

While the authors have mentioned there are no restrictions to making data available, it would be helpful to include information about how data will be made available. "The PLOS Data policy requires authors to make all data underlying the findings described in their manuscript fully available without restriction, with rare exception, at the time of publication. The data should be provided as part of the manuscript or its supporting information, or deposited to a public repository. For example, in addition to summary statistics, the data points behind means, medians and variance measures should be available. If there are restrictions on publicly sharing data—e.g. participant privacy or use of data from a third party—those must be specified."

Reviewer #2: The paper is well written, the protocol is complete and well designed, methods are sound, the goal of the research has potentially a high importance for the clinicians worldwide

However the paper describes a project of research and not the results.

In the criteria of acceptance I found that

"Hypothesis or proposal papers"

are not suitable for publication

I believe that is up to the editor to accept or refuse it

7. PLOS authors have the option to publish the peer review history of their article (what does this mean? ). If published, this will include your full peer review and any attached files.

**Do you want your identity to be public for this peer review?** For information about this choice, including consent withdrawal, please see our Privacy Policy .

Reviewer #1: No

Reviewer #2: **Yes: ** Piergiacomo Calzavara-Pinton

---

## [Author Response · Author response to Decision Letter 0]

27 Dec 2024

Authors’ comments: Thank you for your message. We have reviewed and revised the manuscript to ensure it fully complies with PLOS ONE's style requirements.

Authors’ comments: Thank you for your observation. We have reviewed the grant information and made the necessary corrections to ensure consistency between the ‘Funding Information’ and ‘Financial Disclosure’ sections. The correct grant number(s) for the award(s) supporting this study have been accurately included in the ‘Funding Information’ section. Please let us know if any further adjustments are required.

Authors’ comments: Thank you for your message. We would like to confirm that all the data from our research will be made publicly available after the manuscript is published. We have registered our study with the Open Science Framework (OSF), a free and open-access repository. This ensures the data will be accessible to the public, per the journal's open data policy. Please let us know if any additional information or clarification is needed.

The DOI is DOI 10.17605/OSF.IO/TWRCN

The page https://osf.io/twrcn/

4. We note that the original protocol that you have uploaded as a Supporting Information file contains an institutional logo. As this logo is likely copyrighted, we ask that you please remove it from this file and upload an updated version upon resubmission.

Authors’ comments: We have removed the institutional logo from the original protocol file and the study protocol IRB. The updated version has been uploaded as a Supporting Information file. Please let us know if any further adjustments are required.

Authors’ comments: No retracted articles have been cited in the manuscript. Please let us know if any further adjustments are required.

Reviewers' comments:

Reviewer's Responses to Questions

Comments to the Author

1. Does the manuscript provide a valid rationale for the proposed study, with clearly identified and justified research questions?

Reviewer #1: Yes

Reviewer #2: Yes

2. Is the protocol technically sound and planned in a manner that will lead to a meaningful outcome and allow testing the stated hypotheses?

Reviewer #1: Yes

Reviewer #2: Yes

3. Is the methodology feasible and described in sufficient detail to allow the work to be replicable?

Reviewer #1: Yes

Reviewer #2: Yes

4. Have the authors described where all data underlying the findings will be made available when the study is complete?

Reviewer #1: No

Reviewer #2: Yes

5. Is the manuscript presented in an intelligible fashion and written in standard English?

Reviewer #1: Yes

Reviewer #2: Yes

6. Review Comments to the Author

You may also provide optional suggestions and comments to authors that they might find helpful in planning their study.

Reviewer #1: Overall, this protocol is well written and clear in the proposed methodology. While similar methodologies have been used previously, testing the reproducibility of previous trials will be meaningful. Additionally, the inclusion of secondary outcomes not previously tested will also be of interest to clinicians and researchers. Areas for minor revision are below.

Authors’ comments: We appreciate the reviewer’s positive feedback regarding the clarity and relevance of our protocol. We have addressed the minor revisions suggested, as detailed below. Please let us know if any further adjustments or clarifications are required.

Line 61 - The statement "Although the annual risk of progression of this type of lesion to squamous cell carcinoma (SCC) is 0.025%- 0.6%, participants with multiple lesions have a risk of up to 20% for the emergence of carcinoma [2]" should be rephrased. As it is written, it may be interpreted that patients with multiple AK have a 20% risk of emergence of SCC from AK.

The statistic of 18.8% was for the number of participants developing squamous cell carcinoma in general, not necessarily for developing squamous cell carcinoma from a pre-existing AK. Additionally, this statistic was measured over a 10 year period.

Authors’ comments: Thank you for your insightful feedback regarding the phrasing of our statement on the risk of progression of actinic keratosis (AK) to squamous cell carcinoma (SCC). We recognize the potential for misinterpretation and appreciate the opportunity to correct it (lines 59-65).

“The statistic of 18.8% was for the number of participants developing squamous cell carcinoma in general measured over 10 years. Also, this study shows that 2,893 patients with AK, followed in this period, were compared with those without AK. The study revealed that patients with AK had more than a fivefold increased risk of developing skin cancer in general. Regarding specific types of skin cancer, the risk was particularly high for squamous cell carcinoma, with patients exhibiting a more than sevenfold increased risk [3]”.

Line 221 - It would be beneficial to explain how areas for curettage will be chosen and to clarify that curettage will be done over actinic keratoses and not over the entire face (unless this is not the case and if so this should be made clear).

Authors’ comments: Thank you for your valuable comment. We would like to clarify that all identified actinic keratosis (AK) lesions on the face will undergo curettage before the photosensitizer is applied. This procedure specifically targets the AK lesions, not the entire facial area. The curettage is meticulously performed to ensure that only the areas with AK are treated, thereby optimizing the effectiveness of the subsequent photodynamic therapy. We have included a more accurate explanation in the manuscript (lines 220-223). Please let us know if any further clarifications are required. Thank you very much for the suggestion.

While the authors have mentioned there are no restrictions to making data available, it would be helpful to include information about how data will be made available. "The PLOS Data policy requires authors to make all data underlying the findings described in their manuscript fully available without restriction, with rare exception, at the time of publication. The data should be provided as part of the manuscript or its supporting information, or deposited to a public repository. For example, in addition to summary statistics, the data points behind means, medians and variance measures should be available. If there are restrictions on publicly sharing data—e.g. participant privacy or use of data from a third party—those must be specified."

Authors' response: Thank you for your suggestion. We confirm that all data underlying the findings described in the manuscript will be made fully available without restriction. The data will be provided as part of the manuscript's supporting information and deposited in the Open Science Framework (OSF), a free and open-access public repository. This includes data points behind means, medians, and variance measures, following the PLOS data policy. The DOI for our OSF registration is DOI 10.17605/OSF.IO/TWRCN. Please let us know if any additional information or clarification is required.

Reviewer #2: The paper is well written, the protocol is complete and well designed, methods are sound, the goal of the research has potentially a high importance for the clinicians worldwide

However the paper describes a project of research and not the results.

In the criteria of acceptance I found that

"Hypothesis or proposal papers"

are not suitable for publication

I believe that is up to the editor to accept or refuse it

Author's response: Thank you for your comment. We would like to clarify that PLOS ONE accepts Study Protocols as part of its submission categories. According to the journal's guidelines, Study Protocols must relate to a research study that has not yet generated results, be submitted before participant recruitment or data collection is complete, and adhere to ethical standards, including prior approval from the relevant ethics body. Our submission fulfils these criteria, as it presents a comprehensive research protocol that outlines a detailed plan for a study that is still in its initial stages, with no results generated. We have also obtained prior ethical approval for the study, ensuring compliance with PLOS ONE's submission guidelines for Study Protocols. We hope this clarification addresses the reviewer's concerns. Please let us know if you require any further information.

7. PLOS authors have the option to publish the peer review history of their article (what does this mean?). If published, this will include your full peer review and any attached files.

Do you want your identity to be public for this peer review? For information about this choice, including consent withdrawal, please see our Privacy Policy.

Reviewer #1: No

Reviewer #2: Yes: Piergiacomo Calzavara-Pinton

---

## [Editor Report · Decision Letter 1]

10 Jan 2025

Evaluation of four regimens of methyl aminolevulinate mediated by red light to treat actinic keratoses: a randomized controlled clinical protocol

PONE-D-24-48146R1

Dear Dr. Horliana,

We’re pleased to inform you that your manuscript has been judged scientifically suitable for publication and will be formally accepted for publication once it meets all outstanding technical requirements.

Kind regards,

Liang Liu

Academic Editor

PLOS ONE
---

## [Editor Report · Acceptance letter]

PONE-D-24-48146R1

PLOS ONE

Dear Dr. Horliana,

I'm pleased to inform you that your manuscript has been deemed suitable for publication in PLOS ONE. Congratulations! Your manuscript is now being handed over to our production team.

Kind regards,

on behalf of

Dr. Liang Liu

Academic Editor

PLOS ONE